# Antagonistic Roles of the Tumor Suppressor miR-210-3p and Oncomucin MUC4 Forming a Negative Feedback Loop in Pancreatic Adenocarcinoma

**DOI:** 10.3390/cancers13246197

**Published:** 2021-12-09

**Authors:** Nihad Boukrout, Mouloud Souidi, Fatima Lahdaoui, Belinda Duchêne, Bernadette Neve, Lucie Coppin, Emmanuelle Leteurtre, Jérôme Torrisani, Isabelle Van Seuningen, Nicolas Jonckheere

**Affiliations:** 1Univ. Lille, CNRS, Inserm, CHU Lille, UMR9020-U1277-CANTHER—Cancer Heterogeneity Plasticity and Resistance to Therapies, F-59000 Lille, France; nihad.boukrout@inserm.fr (N.B.); mouloud.souidi@inserm.fr (M.S.); Fatima.Lahdaoui@inserm.fr (F.L.); belinda.duchene@inserm.fr (B.D.); bernadette.neve@inserm.fr (B.N.); lucie.coppin@inserm.fr (L.C.); Emmanuelle.Leteurtre@chru-lille.fr (E.L.); isabelle.vanseuningen@inserm.fr (I.V.S.); 2Université de Toulouse, INSERM, Université Toulouse III-Paul Sabatier, Centre de Recherches en Cancérologie de Toulouse, F-31037 Toulouse, France; jerome.torrisani@inserm.fr

**Keywords:** pancreatic cancer, MUC4, miR-210-3p, anti-tumoral miR

## Abstract

**Simple Summary:**

We aimed at characterizing microRNAs activated downstream of MUC4-associated signaling in pancreatic adenocarcinoma. We investigated the MUC4-miR-210-3p reciprocal regulation and deciphered miR-210-3p biological roles in vitro and in vivo. We showed a MUC4-miR-210-3p negative feedback loop that involves NF-κB in PDAC-derived cells and the miR-210-3p anti-tumoral functions, suggesting a complex balance between antagonistic pro-oncogenic functions of the oncomucin MUC4 and anti-tumoral roles of the miR-210-3p.

**Abstract:**

Background: Pancreatic adenocarcinoma (PDAC) is a deadly cancer with an extremely poor prognosis. MUC4 membrane-bound mucin is neoexpressed in early pancreatic neoplastic lesions and is associated with PDAC progression and chemoresistance. In cancers, microRNAs (miRNAs, small noncoding RNAs) are crucial regulators of carcinogenesis, chemotherapy response and even metastatic processes. In this study, we aimed at identifying and characterizing miRNAs activated downstream of MUC4-associated signaling in pancreatic adenocarcinoma. MiRnome analysis comparing MUC4-KD versus Mock cancer cells showed that MUC4 inhibition impaired miR-210-3p expression. Therefore, we aimed to better understand the miR-210-3p biological roles. Methods: miR-210-3p expression level was analyzed by RT-qPCR in PDAC-derived cell lines (PANC89 Mock and MUC4-KD, PANC-1 and MiaPACA-2), as well as in mice and patients tissues. The MUC4-miR-210-3p regulation was investigated using luciferase reporter construct and chromatin immunoprecipitation experiments. Stable cell lines expressing miR-210-3p or anti-miR-210-3p were established using CRISPR/Cas9 technology or lentiviral transduction. We evaluated the biological activity of miR-210-3p in vitro by measuring cell proliferation and migration and in vivo using a model of subcutaneous xenograft. Results: miR-210-3p expression is correlated with MUC4 expression in PDAC-derived cells and human samples, and in pancreatic PanIN lesions of Pdx1-Cre; LstopL-KrasG12D mice. MUC4 enhances miR-210-3p expression levels via alteration of the NF-κB signaling pathway. Chromatin immunoprecipitation experiments showed p50 NF-κB subunit binding on miR-210-3p promoter regions. We established a reciprocal regulation since miR-210-3p repressed MUC4 expression via its 3′-UTR. MiR-210-3p transient transfection of PANC89, PANC-1 and MiaPACA-2 cells led to a decrease in cell proliferation and migration. These biological effects were validated in cells overexpressing or knocked-down for miR-210-3p. Finally, we showed that miR-210-3p inhibits pancreatic tumor growth and proliferation in vivo. Conclusion: We identified a MUC4-miR-210-3p negative feedback loop in early-onset PDAC, but also revealed new functions of miR-210-3p in both in vitro and in vivo proliferation and migration of pancreatic cancer cells, suggesting a complex balance between MUC4 pro-oncogenic roles and miR-210-3p anti-tumoral effects.

## 1. Introduction

Pancreatic ductal adenocarcinoma (PDAC) is an extremely aggressive and highly drug-resistant cancer. At the time of diagnosis, more than 80% of PDACs are already locally advanced or metastatic. In addition, the 5-year survival rate of patients with local or metastatic PDAC treated with surgery or palliative chemotherapy, respectively, remains very low (<10%) [1,2]. Owing to its dramatic prognosis and the failure of conventional therapies, PDAC is the third leading cause of cancer death in the United States and it is predicted to be the second cause by 2030 [3]. Understanding PDAC carcinogenesis mechanisms and identifying the main factors of its invasiveness are crucial for patient care improvement and the development of new therapeutic approaches. Pancreatic precancerous lesions are the starting point of PDAC development. The most common lesions are pancreatic intraepithelial neoplasia (PanIN) [4]. Their initiation and progression are frequently associated with an abnormal synthesis of mucins and especially with neoexpression of mucin-4 (MUC4), which is undetectable in the normal pancreas [5]. Furthermore, MUC4 neoexpression is positively correlated with PDAC progression and aggressiveness [6].

MUC4 is a high molecular weight *O*-glycoprotein (up to 930 kDa). This heterodimeric membrane-bound protein is composed of two subunits: MUC4α and MUC4β. The extracellular mucin-type subunit, MUC4α, represents a typical hyper-glycosylated region. The membrane-bound EGF-like subunit, MUC4β, contains EGF-like domains and is implicated in receptor–ligand interactions with the transmembrane growth factor receptor ErbB2/HER2 [7]. MUC4 alone and the MUC4–ErbB2 complex contribute to pancreatic tumorigenesis via the regulation of cell signaling pathways related to cell proliferation, survival, motility [8] and chemotherapy metabolism [9]. Several genes activated downstream of MUC4-regulated cell signaling have been investigated. Indeed, Skrypek et al. demonstrated a transcriptional activation of the human nucleoside transporters hCNT1 via the MUC4-regulated NF-κB pathway in pancreatic cancer cells [9]. However, it remains unknown whether MUC4 is implicated in microRNA (miRNA) gene regulation and thereby controls epigenetic mechanisms.

MiRNAs are evolutionarily conserved small non-coding RNA (18–25 nt) that are negative post-transcriptional regulators of gene expression via the interaction with 3′ UTR of targeted mRNAs inducing either their cleavage or translation inhibition [10]. MiRNAs have been demonstrated as regulators of biological processes and cellular homeostasis during the initiation and the progression of many diseases, including cancers. Oncogenic or tumor-suppressor effects have been described depending on their expression level, mRNA targets and environmental conditions. Because of their high stability, miRNAs can be isolated from body fluids, paraffin-embedded or fresh tissue samples and measured by qPCR [11,12]. Altogether, miRNAs seem to be interesting therapeutic targets and potential biomarkers.

MiR-210 is one of the most frequently dysregulated miRNAs in PDAC [13]. It is highly expressed in pancreatic cancer tissues [14,15], as well as in the patients’ plasma [16,17]. miR-210 overexpression in PDAC tissue samples was shown as a predictor of poor outcome [15], whereas an increased plasma level was linked to better patient survival [17]. Moreover, plasma miR-210 concentration combined with other miRNA levels has a good diagnostic value for PDAC [18,19]. However, implication of miR-210 in pancreatic carcinogenesis remains unclear. Moreover, some studies described miRNAs such as miR-219-1-3p or miR-150 as negative regulators of MUC4 in pancreatic cancer cells [20,21]. However, it remains unknown whether miR-210-3p is a MUC4 regulator.

In this manuscript, we demonstrated for the first time a reciprocal regulation between a mucin and a miRNA. Indeed, we identified and characterized a MUC4-miR-210-3p negative feedback loop in PDAC, but we also elucidated the miR-210-3p roles in vitro and in vivo on proliferation and migration of pancreatic cancer cells.

## 2. Results

### 2.1. MiR-210-3p Is Overexpressed in PDAC

In order to identify miRNAs potentially regulated by MUC4, we analyzed a miRnome dataset based on MUC4 knock-down (MUC4-KD) 647-V cancer cells (unpublished data, GSE137130) and observed a decrease in miR-210 relative expression level in MUC4-KD compared to Mock control cells by reverse transcription quantitative polymerase chain reaction (RTqPCR, *p* < 0.001) (Appendix A).

Based on these observations, we then investigated miR-210-3p study in pancreatic cancer samples. MiR-210-3p expression was evaluated by RT-qPCR in both PDAC human tissues and pancreatic cancer cell lines. We found a significant miR-210-3p overexpression in nine paired human PDAC tissues compared to their corresponding non-tumoral adjacent tissues (4.7 ± 3.8-fold increase, *p* = 0.0112, Figure 1A). We obtained a similar result in the GSE41369 PDAC dataset (4.2 ± 4.9-fold increase, *p* = 0.0320, Figure 1B). In addition, miR-210-3p overexpression was found in PDAC cell lines PANC89 and PANC-1 compared with normal human pancreatic ductal HPDE cells (Figure 1C). Interestingly, the highest miR-210-3p expression level fold change (5.93 ± 0.73 fold change) was observed in PANC89 cells expressing MUC4 compared to PANC-1 and MIA PaCa-2 cells that do not express MUC4 (1.8 ± 0.05 fold change, and 0.43 ± 0.02-fold, respectively, Figure 1C,D).

We also determined that MUC4 down-expression in PANC89 cells by shRNA (Figure 1F) leads to a significant decrease in miR-210-3p relative expression level compared to the corresponding Mock control cells (38.10 ± 4.15% decrease, *p* = 0.0008) (Figure 1E). We verified the decrease in MUC4 expression following stable shMUC4 transfection (Figure 1F).

Altogether, our results show a miR-210-3p overexpression in PDAC samples and suggest an association between MUC4 and miR-210-3p expression in PDAC-derived cells.

### 2.2. MiR-210-3p Expression Is Positively Correlated with Muc4 Expression during Pancreatic Carcinogenesis

In order to establish a link between MUC4 and miR-210-3p in early-stage PDAC, we evaluated miR-210-3p expression level by RT-qPCR and Muc4 immuno-staining scores by immunohistochemistry (IHC) in Pdx1-Cre; K-rasG12D (KC) transgenic mouse model of pancreatic cancer. The KC mouse model harbors PanIN formation that increases in size and number over time (3 to 12 months) (Figure 2A). As expected, we did not observe any PanIN in Pdx1-Cre; K-rasWT (WT) control mice. Moreover, caerulein intraperitoneal injections enhanced a loss of normal exocrine histology and promoted PanIN progression as previously described by Guerra et al. (2007) [22]. We confirmed the Muc4 sustained expression in the cell membrane in PanIN lesions at every age, as previously observed in [23], and also in pancreas from caerulein-treated KC mice. In contrast to miR-210-3p, we did not observe a Muc4 basal expression in WT mice normal pancreas. Muc4 immuno-staining scores showed a statistically significant increase in Muc4 expression at 12 months old and in caerulein-treated mice compared to WT mice (Figure 2B). Interestingly, as observed in MUC4 IHC, RTqPCR analysis demonstrated that miR-210-3p expression is also significantly increased at 12 months old and in caerulein-treated mice (Figure 2C). Muc4 immuno-staining scores and miR-210-3p relative expression level were positively correlated with both Pearson’s and Spearman’s r higher than 0.7 (Figure 2D). In addition, we also noticed a positive correlation between *MUC4* and miR-210-3p RNA relative expression levels in patients from a TCGA PAAD public dataset (Spearman *r* = 0.2719, *p* < 0.001) (Appendix A).

Altogether, our results highlight a positive correlation between MUC4 and miR-210-3p expression levels in mice and patients’ PDAC samples and suggest a potential MUC4 implication in miR-210-3p regulation.

### 2.3. MUC4 Regulates miR-210 Expression at the Transcriptional Level

We previously demonstrated MUC4 involvement in transcriptional gene regulation via NF-κB pathway modulation [9]. In this study, we investigated whether miR-210-3p expression level was also subject to this MUC4 regulatory mechanism via the NF-κB signaling pathway. We first confirmed the MUC4 impact on NF-κB pathway activity in PANC89. Indeed, we showed that MUC4-KD in PANC89 cells induce a 57.4 ± 9.9% significant decrease in relative luciferase activity of the kB-Luc synthetic promoter compared to Mock cells (*p* < 0.0001, Figure 3A). We then performed transient NF-κB knock-down in PANC89 cells and validated NF-κB p50 siRNA efficacy by Western blot analysis showing a strong decrease in p50 NF-κB subunit expression (41%, *p* < 0.001, Figure 3B). We observed a significant decrease in miR-210-3p expression level in PANC89 p50 NF-κB knock-down cells compared to non-targeting (NT) siRNA control cells (62.7 ± 3.0% decrease, *p* < 0.0001, Figure 3C). Using the Eukaryotic Promoter Database (https://epd.epfl.ch//index.php, accessed on 8 January 2019), we analyzed the miR-210 promoter and identified numerous κB putative binding sites in silico. We then investigated p50 NF-κB subunit direct binding on the miR-210 promoter following chromatin immunoprecipitation experiments. We observed a p50 NF-κB binding enrichment compared to IgG negative control on four miR-210 promoter regions (P1–P4), encompassing seven different κB p50 binding sites tested in PANC89 cells (Figure 3D,E). Interestingly, we observed that these NF-κB p50-miR-210 promoter interactions were lost in PANC89 MUC4-KD cells.

Altogether, these results indicate that MUC4 transcriptionally regulates miR-210-3p expression via NF-κB pathway activation and direct p50 binding on the miR-210 promoter.

### 2.4. MiR-210-3p Represses MUC4 Expression in PDAC-Derived Cells

After demonstrating that MUC4 mediates miR-210 expression regulation (Figure 3), we aimed at determining the existence of MUC4 and miR-210-3p reciprocal regulation. Using the MicroCosm Targets miRNA database, (European Bioinformatic institute), we identified in silico three putative miR-210-3p binding sites in MUC4 3′-UTR at positions 219–240 (site #1), 159–182 (site #2) and 248–268 (site #3), respectively (Figure 4A). In order to study MUC4 regulation by miR-210-3p, we generated stable PANC89 AAVS1 cell lines expressing miR-210-3p, anti-miR-210-3p or their corresponding controls miR-scramble and anti-miR-control, respectively. We validated that our cell models harbored an increased miR-210-3p relative expression level in AAVS1-miR-210-3p cells (3-fold) and a decreased relative expression level in AAVS1-anti-miR-210-3p cells (54% decrease) compared to their corresponding controls, miR-scramble and anti-miR-control, respectively (Figure 4B). Western blot analysis showed a strong MUC4 expression inhibition in PANC89 AAVS1-miR-210-3p cells (91% decrease, *p* < 0.001, Figure 4C) compared to AAVS1-miR-scramble cells. On the contrary, anti-miR-210-3p induced an increase in MUC4 expression level (1.9-fold, *p* < 0.01) compared to AAVS1- anti-miR-control cells. Finally, we investigated whether miR-210-3p regulates MUC4 expression through its 3′-UTR. Therefore, we performed a co-transfection of MUC4 3′-UTR-luciferase reporter construct and miR-210-3p in PANC-1 cells to prevent interference or sponge effect of endogenous MUC4 mRNA. We observed a significant reduction in relative luciferase activity in PANC-1 cells overexpressing miR-210-3p compared to PANC-1 miR-scramble cells (34.4 ± 9.3% decreases, *p* = 0.0215, Figure 4D).

Altogether, these results show that miR-210-3p represses MUC4 expression via its 3′-UTR.

### 2.5. MiR-210-3p Inhibits PDAC-Derived Cell Proliferation and Migration In Vitro

We further investigated the miR-210-3p biological roles in vitro in PANC89 (expressing MUC4), PANC-1 and MIA PaCa-2 (not expressing MUC4) pancreatic cancer cells by performing transient and stable transfections. MiR-210-3p transfection efficacy was evaluated by RT-qPCR (Appendix A). We first assessed MTT assays to evaluate global miR-210-3p effects on cell viability. We observed a significant decrease in cell viability in all transient and stable miR-210-3p overexpressing cells compared to miR-scramble overexpressing cells (20–40% decrease *p* < 0.001 Figure 5A). On the contrary, anti-miR-210-3p induced a significant increase in cell viability in PANC-1 and PANC-89 AAVS1 anti-miR-210-3p cells compared to AAVS1 anti-miR-control cells (63.4 ± 0.02 and 72.48 ± 0.02% increase, respectively, *p* < 0.001 Figure 6A). We then studied miR-210-3p’s roles in cell proliferation using Incucyte™ technology. Cell confluency was significantly decreased in miR-210-3p overexpressing cells while miR-210-3p inhibition significantly enhanced PANC89 and PANC-1 cell proliferation compared to control conditions (Figure 5B and Figure 6B). Finally, we performed wound healing assays in order to investigate miR-210-3p’s impact on cell-migration in pancreatic cancer cells. We showed that miR-210-3p overexpression significantly reduces the cell-migration compared to miR-scramble cells. Moreover, anti-miR-210-3p enhances cell-migration in PANC89 and PANC-1 cells compared to anti-miR-control cells (Figure 5C,D and Figure 6C,D). Interestingly, we noticed that the highest miR-210-3p effects on cell viability, proliferation and migration processes were observed in MUC4-expressing PANC89 cells compared to PANC-1 and MIA PaCa-2 cells.

Altogether, these results highlight the anti-tumoral roles of miR-210-3p in pancreatic cancer cells.

### 2.6. MiR-210-3p Inhibits Pancreatic Tumor Growth In Vivo

In order to investigate miR-210-3p’s in vivo functions, we used Capan-1 PDAC cells that were previously transfected with a lentivirus encoding miR-210-3p (LV-miR-210-3p and LV-miR-neg control cells). First, we validated a strongest miR-210-3p overexpression using RTqPCR analysis (13.8 ± 1.2-fold change, *p* < 0.001, Figure 7A). We confirmed the in vitro miR-210-3p anti-proliferation effect in Capan-1 LV-miR-210-3p compared to LV-miR-neg cells (38.1% decrease, *p* < 0.001) (Figure 7B) as previously demonstrated in Figure 5 and Figure 6. These cells were then subcutaneously xenografted in SCID mice and tumor growth was evaluated for 51 days. We observed that LV-miR-210-3p Capan-1 tumors were significantly smaller than LV-miR-neg control tumors (42% decrease, *p* < 0.05) at 39 days and the difference was sustained until mice were euthanized (Figure 7C). Tumors were extracted and Ki67 index was analyzed by immuno-staining (Figure 7D,E). We observed a statistically significant decrease in the Ki-67^+^ cell percentage in tumors from LV-miR-210-3p (66.6 ± 18.9 decrease, *p* = 0.0245) compared to miR-neg control tumors. Moreover, Western blot analysis of Capan-1 LV-miR-210-3p cells showed a decrease in proliferation-associated protein levels, such as S474 phospho-Akt, constitutive Akt and Cyclin D1 compared to Capan-1 LV-miR-neg control cells (Figure 7F and Appendix A).

Altogether, these results demonstrate that the miR-210-3p ectopic over-expression inhibits Capan-1 cell proliferation and decreases tumor growth in vivo.

## 3. Discussion

In the present manuscript, we observed that both MUC4 and miR-210 are overexpressed in PDAC. Interestingly, inhibition of MUC4 expression leads to a decrease in miR-210-3p relative level via the alteration of the NF-κB signaling pathway. Moreover, we also showed that miR-210-3p represses MUC4 expression via its 3′-UTR, suggesting a negative feedback regulation loop. Finally, we characterized miR-210-3p roles in PDAC using in vitro and in vivo approaches and showed that it acts as an anti-tumor miRNA.

MUC4 is a key actor in pancreatic carcinogenesis [24,25]. Since its neoexpression in PanIN’s earlier stages, MUC4 contributes to progression and aggressiveness of PDAC. Several in vitro and in vivo studies demonstrated that MUC4 modulates pancreatic tumor growth, cell proliferation, invasion and apoptosis [8,9,26,27]. Nevertheless, the underlying mechanisms associated with MUC4 expression remain to be fully deciphered. In this study, we identified miR-210-3p as a MUC4-regulated microRNA and deciphered the NF-kB mechanism involved in this regulation. Finally, we demonstrated the miR-210-3p effects on the biological properties of pancreatic cancer cells. MiR-210-3p was initially described in ovarian cancer as a crucial player in tumor onset and a key regulator of the hypoxia response [28]. MiR-210-3p is a predominant hypoxia HIF1α-inducible microRNA in a broad spectrum of cancer types, including pancreatic cancer. In addition to its well-described role as a hypoxic regulator, miR-210-3p is also expressed under normoxia conditions and modulates tumor initiation [29]. In this study, we aimed at investigating MUC4-miR-210 roles in earlier stages of pancreatic tumor formation before the hypoxia response activation. We identified and characterized the MUC4-miR-210-3p feedback regulation loop in PDAC cells under normoxia conditions and showed that miR-210-3p inhibits pancreatic cancer cell proliferation and migration. We also observed that MUC4 is induced in hypoxic conditions and that miR-210-3p relative level is reduced when MUC4 expression is inhibited (MUC4-KD cells) compared to control Mock cells (Jonckheere, unpublished results).

In PDAC, both MUC4 and miR-210-3p are up-regulated [5,14,15,25] and associated with poor prognosis [15,30]. In this study, we revealed a positive correlation between Muc4 expression and miR-210-3p during PanIN initiation and progression. Moreover, the inhibition of MUC4 in PDAC cells induces a significant decrease in miR-210-3p expression levels. Our result suggests a potential implication of MUC4 in miR-210-3p transcriptional regulation. Indeed, it is now established that MUC4 is able to modulate gene expression through cell signaling activation as we observed a significant decrease in NF-κB activity in MUC4-KD PANC89 pancreatic cancer cells. For instance, we previously showed that MUC4 induced hCNT1 upregulation in PDAC cells via NF-κB pathway modulation [9]. NF-κB is a family of dimeric transcription factors central to inflammatory responses, immunity, cellular differentiation, proliferation and survival in multicellular organisms [31]. The NF-κB network dysregulation has been implicated in a wide range of diseases, including cancers. In PDAC, it promotes tumor progression by regulating genes implicated in proliferation, angiogenesis and survival. The NF-κB pathway is well known to be constitutively activated in PDAC compared to normal pancreas [32]. The regulatory network formed by transcription factors and miRNA has been extensively studied. The mir-210 gene is located in an intron of a noncoding RNA, miR210HG, located on chromosome 11p15.5 (NCBI, gene, 406992). A functional NF-κB p50-binding site was identified and acts as transcriptional activator of miR-210 expression in pre-eclampsia disease [33]. In our work, we demonstrated that NF-κB inhibition in PANC89 pancreatic cancer cells using siNF-κB p50 significantly decreased the miR-210-3p relative level. Based on this result, we investigated NF-κB p50 implication on miR-210-3p transcriptional regulation under MUC4 control in pancreatic cancer cells. Mapping of a 2-kb core promoter region immediately upstream of the miR-210-3p stem-loop structure allowed us to identify several putative κB binding sites. Structural and functional studies revealed that NF-κB p50 can physically interact with the miR-210 promoter and transactivate it in PDAC-derived cells. Interestingly, we demonstrated that the MUC4-KD abolished NF-κB p50-miR-210 promoter interactions via NF-κB activity inhibition. We previously showed NF-κB subunit accumulation in the cytoplasmic fraction of MUC4-KD cancer cells [9].

In silico analysis of MUC4 3′-UTR allowed us to identify three miR-210-3p binding sites. This result suggests that MUC4 is a potential miR-210 target. Complex regulation of MUC4 has been demonstrated, as MUC4 is regulated (1) at the epigenetic level via DNA methylation and histone modifications [34], (2) at the transcriptional level via the epidermal growth factor [35], transforming growth factor β [36,37] and Kras-activated MAPK and NF-κB pathways [23] and (3) at the post-transcriptional level via miRNAs. Indeed, in PDAC-derived cells, MUC4 is a functional target of miR-150 [21] and miR-219-1-3p [20]. Similarly, in this report, we demonstrate that miR-210-3p is a MUC4 regulator. We demonstrated that miR-210-3p inhibits MUC4 expression at the post-transcriptional level via the MUC4 3′-UTR. In this study we showed that MUC4 and miR-210-3p regulate each other. MiR-210-3p transcriptional activation by MUC4 appears to be sustained during pancreatic carcinogenesis as their expressions are positively correlated and both MUC4 and miR-210-3p are overexpressed in PDAC cells and tissues.

Conversely, we hypothesize that MUC4 escapes miR-210-3p regulation during late pancreatic cancer stages since MUC4 is aberrantly upregulated in advanced PDAC even in the presence of high miR-210-3p expression levels. Accordingly, we observed similar findings in KC mice, which harbored strong MUC4 immuno-staining and high levels of miR-210-3p, suggesting that cellular mechanisms promoting MUC4 expression can overcome miR-210-3p-induced repression. This could be due to altered expression of other miR-210 targets that trap the miRNA in other regulons. We therefore suggest that the MUC4–miR-210 negative feedback loop is exclusively effective in earlier stages of pancreatic cancer in order to maintain cell homeostasis and inhibit tumor initiation under normoxic conditions (Figure 8).

Although miR-210 tissue expression is inversely correlated to survival in PDAC patients [15], we showed that miR-210-3p ectopic expression inhibits in vitro proliferation and migration of pancreatic cancer cells and tumor growth in a xenograft model. Several studies show controversial miR-210-3p effects on pancreatic cancer cells biological processes. It is well known that miRNAs had different roles depending on their origin, spatiotemporal expression, environmental stimuli and cell types. Indeed, several studies demonstrated an oncogenic miR-210-3p effect. Yang and colleagues suggested that miR-210-3p in exosomes derived from gemcitabine-resistant pancreatic cancer stem cells induce drug resistance in gemcitabine-sensitive pancreatic cancer [38]. Moreover, miR-210 mediates the occurrence of epithelial–mesenchymal transition (EMT) of pancreatic cancer cells under hypoxia [39]. In contrast, other studies suggested a miR-210-3p tumor-suppressing effect as it inhibits pancreatic cell proliferation [40] and represses the initiation of tumor growth under normoxic conditions [29]. The overall balance between miR-210-3p pro- and anti-tumorigenic effects may depend on its spatiotemporal expression (center or periphery of tumor, early or late phases of cancer) and environmental stimuli (such as hypoxia, chemotherapy).

In summary, we show for the first time the existence of a reciprocal regulation loop between MUC4 and miR-210-3p. Moreover, our findings indicate that miR-210-3p could be a good anti-tumor candidate by inhibiting both MUC4 expression and tumor initiation.

## 4. Materials and Methods

### 4.1. Human Pancreatic Ductal Tumor Sample

A total of nine paired human PDAC tissues and their corresponding adjacent normal tissues were collected from PDAC patients in Lille University Hospital (Lille, France) [41]. Every patient signed an informed consent form of non-opposition to research use of a biological sample. A part of surgically resected samples was immediately fixed in formaldehyde and embedded in paraffin. All patients were naive of any chemotherapy prior to surgery.

### 4.2. Pdx1-Cre; LSL-KrasG12D Mouse Model

Pdx1-Cre (C57Bl/6 background) and LStopL-KrasG12D (C57Bl/6 background) transgenic mice were previously described in [23]. LSL-KrasG12D and Pdx1-Cre mice were maintained as heterozygous lines and crossed to obtain Pdx1-Cre; LSL-KrasG12D (KC). After sacrifice and dissection, pancreas from 3-, 6-, 9- and 12-month old KC and WT control mice were fixed and embedded in paraffin.

Intraperitoneal injections of 37.5 µg/mL caerulein solution were performed on 6-month-old KC and WT control mice following two processing steps. First, an acute treatment with an injection every hour for 6 h (1st day) followed by a chronic treatment with an injection every day (5 days a week) for 59 days. At the end of the protocol, pancreas were dissected, fixed and embedded in paraffin. All procedures were in accordance with the guideline of Animal Care Committee (#00422.02).

### 4.3. Cell Lines and Culture Conditions

PANC89 pancreatic cancer cells and 647-V bladder cancer cells were obtained from Dr FX Real (CNIO, Madrid, Spain). MIA PaCa-2 (ATCC^®^ CRL-1420™) and PANC-1 (ATCC^®^ CRL-1469™) pancreatic cancer cells were purchased from the American Type Culture Collection (ATCC). MUC4 knocked-down (MUC4-KD) cells were obtained as described previously [9]. Cells were cultured in RPMI (PANC-89, Capan-1, 647-V) or DMEM (MIA PaCa-2, PANC-1) media containing, respectively, 15% (PANC-89 and Capan-1) and 10% (other cells) of heat-inactivated Fetal Bovine Serum and supplemented with 2mM L-glutamine and 1% penicillin-streptomycin solution at 37 °C in a 5% CO_2_ humidified atmosphere. These cell lines were authenticated according to the procedures recommended by the ATCC Institute.

### 4.4. Gene Expression Omnibus Microarray

Public pancreatic cancer microarray (GSE41369) was analyzed from the NCBI Gene Expression Omnibus (GEO) database (https://www.ncbi.nlm.nih.gov/geo/, accessed on 8 December 2018). The mir-210 expression profile was established using nine tumors and adjacent non-tumor tissues from PDAC cases. Data were analyzed using GEO2R software.

### 4.5. Cell Transient Transfection

Transient miR-210-3p overexpression was performed using 30 nM of pre-miR-210-3p and siPORTNeoFX transfection reagent (Ambion, Thermo Fisher Scientific, Illkirch-Graffenstaden, France). Transient NF-κB knockdown was performed using 5 µM siRNA (NF-κB1) from Dharmacon (Thermo Fisher Scientific, Illkirch-Graffenstaden, France). The manufacturer’s instructions were followed in both protocols. Controls were performed using pre-miR-scramble and non-targeting siRNA (NT). Transfection efficiency was evaluated, respectively, by RT-qPCR and Western blot analysis. Co-transfection of 1 µg of pGL3-MUC4-3′UTR luciferase reporter plasmid obtained as described previously [20] and 30 nM of pre-miR-210-3p was performed using Lipofectamine 3000^TM^ transfection reagent (Thermo Fisher Scientific, Illkirch-Graffenstaden, France). Transfection of κB-Luc synthetic promoter containing three κB-binding sites was performed with Lipofectamine 3000^TM^ transfection reagent (Thermo Fisher Scientific, Illkirch-Graffenstaden, France). A luciferase reporter assay was performed 48 h post transfection.

### 4.6. Luciferase Reporter Assay

Relative luciferase activity was evaluated using a Mithras Microplate Reader LB 940 (Berthold Technologies, Bad Wildbad, Germany) on cell lysates extracted using Reagent Lysis^®^ Buffer (Promega, Madison, WI, USA). Protein concentration was determined using the bicinchoninic acid method. Relative luciferase activity was normalized (/total protein concentrations) and expressed as a percentage of fold activation compared to control conditions. Each experiment was performed in triplicate.

### 4.7. Establishment of miR-210 Stable Cell Lines by CRISPR/Cas9 Genome Editing

PANC89 and PANC-1 AAVS1 miR-210-3p and anti-miR-210-3p and their corresponding controls AAVS1 miR-scramble and miR-control stable cell lines were established using the genome editing strategy. The coding DNAs for miR-210-3p, anti-miR-210-3p, miR-scramble and miR-control (Table 1) were cloned into pAAVS1-shRNA expression vector (#82697 Addgene, Watertown, MA, USA), under the U6 promoter and flanked by Adeno-Associated Virus Integration Site 1 (AAVS1) homology arms sequences. The expression vector was co-transfected with pCas9_GFP (#44719 Addgene, Watertown, MA, USA) and AAVS1 guide RNA (#41824 Addgene, Watertown, MA, USA) using Lipofectamine 3000 ™ Transfection Reagent (Thermo Fisher Scientific, Illkirch-Graffenstaden, France) according to the manufacturer’s instructions. Positive PANC89 and PANC-1 cells were selected with 2 µg/mL of puromicin (Promega, Madison, WI, USA) chronic treatment. Genotyping experiments by AAVS1 PCR were then performed to validate the insertion of the constructs. Expression of mature miR-210-3p was quantified by RT–qPCR as described below. pCas9_GFP was a gift from Dr K. Musunuru (Addgene plasmid #44719; http://n2t.net/addgene:44719; RRID: Addgene_44719, Watertown, MA, USA) [42]. pAAVS1-shRNA was a gift from Dr A. Mullen (Addgene plasmid #82697; http://n2t.net/addgene:82697; RRID: Addgene_82697, Watertown, MA, USA) [43]. gRNA_Cloning Vector was a gift from Dr G. Church (Addgene plasmid #41824; http://n2t.net/addgene:41824; RRID:Addgene_41824, Watertown, MA, USA) [44]. Capan-1 LV-miR-neg and Capan-1 LV-miR-210 stable cell lines were obtained from Dr J. Torrisani (CRCT, Toulouse, France) as previously described in [20].

### 4.8. Chromatin Immunoprecipitation (ChIP)

Crosslinking of chromatin proteins to DNA was performed for 10 min at RT by adding formaldehyde drop-wise directly to the cell-culture medium to a final concentration of 1%. The reaction was stopped by the addition of 0.125 M glycine for 5 min. Cells were washed with 1× PBS, scrapped, and collected by centrifugation. Chromatin was prepared from nuclei purified by two successive extraction steps at 4 °C for 10 min, with 50 mM Hepes/KOH (pH 7.5); 140 mM NaCl; 1 mM EDTA; 10% Glycerol; 0.5% NP-40; 0.25% Triton X-100] and [200 mM NaCl; 1 mM EDTA; 0.5 mM EGTA; 10 mM Tris (pH 8.0). Nuclei were resuspended in 50 mM Tris (pH 8.0); 0.1% SDS; 1% NP-40; 0.1% Na-Deoxycholate; 10 mM EDTA; 150 mM NaCl, supplemented with protease inhibitor cocktail (Sigma, P8340) and sonicated with Bioruptor Power-up (Diagenode, Toyama, Japan), yielding genomic DNA fragments with a bulk size of 150–300 bp. Chromatin was recovered by centrifugation at 14,000× g for 10 min at 4 °C and immunoprecipitation carried out overnight at 4 °C with specific antibodies directed against NF-κB p50 transcription factor (N-19, sc-1191) and the non-specific IgG control (bovine anti-goat IgG-HRP, sc-2350). Immune complexes were recovered by adding protein G-coupled magnetic beads and incubated for 2 h at 4 C. Beads were washed as follows: low salt buffer (0.1% SDS; 1% Triton X-100; 2 mM EDTA; 20 mM Tris (pH 8.0); 150 mM NaCl) (×2), high salt buffer (0.1% SDS; 1% Triton X-100; 2 mM EDTA; 20 mM Tris (pH 8.0); 500 mM NaCl) (×2), LiCl wash buffer (10 mM Tris (pH 8.0); 1% Na-deoxycholate; 1% NP-40, 250 mM LiCl; 1 mM EDTA] (once), and TE supplemented with 50 mM NaCl) (×2). Elution of cross-linking complexes was performed at 65°C using elution buffer (5 mM Tris–HCl, pH 8.0, 25 mM EDTA, 10% (*v*/*v*) SDS). After reversion of cross-linking with 5 M of NaCl and digestion of chromatin-associated proteins with protease-K (Qiagen), DNA was purified using NucleoSpin Gel and PCR Clean-up kit (Macherey-Nagel™, Allentown, PA, USA). RT-PCR analyses were conducted on 1 μL of chromatin using SsoFast Evagreen Supermix kit (Bio-Rad, Marnes La coquette, France) and CFX96 real-time PCR system (Bio-Rad, Marnes La coquette, France). Primers are listed in Table 2. Chromatin enrichment was normalized to input samples.

### 4.9. RNA Isolation and Quantitative Reverse Transcription-Polymerase Chain Reaction (RT-qPCR)

Formalin-fixed paraffin-embedded (FFPE) sample RNAs (from mice and human cells) were purified, respectively, using NucleoSpin^®^ miRNA (Macherey-Nagel™, Allentown, PA, USA) and RecoverAll™ Total Nucleic Acid Isolation Kit (Ambion, Life Technologies, Tokyo, Japan). MiRNA was measured by RT-qPCR using a TaqMan^®^ MicroRNA Assays protocol (Thermo Fisher Scientific, Illkirch-Graffenstaden, France) and CFX96 ™ Real-Time System (Bio-Rad, Marnes La Coquette, France). Briefly, 5 ng of total RNA was reverse transcribed using a TaqMan MicroRNA Reverse Transcription kit, including Taqman microRNA primers specific for miR-210-3p (hsa-miR-210-3p, 000512), RNU48 (001006) and snoRNA202 (001232) from Thermo Fisher Scientific, following manufacturer’s instructions. The qPCR was carried out using TaqMan Gene Expression Master Mix II with Taqman microRNA specific primers using recommended PCR cycling conditions. Expression levels were normalized using RNU48 (human cells and tissues), or snoRNA202 (mice samples) and were measured using the 2^−ΔΔCt^ method.

### 4.10. Immunoblotting

Total protein extraction and Western blotting analysis were carried out as described before [45,46]. After Western blotting, the nitrocellulose membrane (0.2 μm, Schleicher et Shüll, Life Technologies, Tokyo, Japan) was incubated with Akt (9272, Cell Signaling Technology, 1/500, Danvers, MA, USA), Cyclin D1 (M-21, sc 718, 1/500), NF-κB p50 (clone H-119, 1/500), and MUC4 (clone 8G7, 1/200) from Santa Cruz Biotechnology Inc and β-actin (AC-15, 1/10000, Sigma-Aldrich, Saint-Quentin-Fallavier, France). Antibodies were diluted in 5% (*w*/*v*) non-fat dry milk in Tris-buffered saline (Tween-20). MUC4 and β-actin antibodies were diluted in Tris-buffered saline Tween-20 and incubated overnight at 4 °C. Peroxidase-conjugated secondary antibodies were used (Sigma-Aldrich, Saint-Quentin-Fallavier, France). Immunoreactive bands were visualized with the LAS4000 device (Fujifilm, Courbevoie, France) using the Super Signal^®^ West Pico chemoluminescent substrate (Thermo Scientific, Illkirch-Graffenstaden, France). Band density was quantified with the image analysis software ImageJ and represented as histograms.

### 4.11. Analysis of Cell Properties

MTT assay: Cell viability was measured using MTT assay. Briefly, 10^4^ cells were cultured over 96 h. MTT (Sigma-Aldrich, 0.5 ng/Ml, Saint-Quentin-Fallavier, France) was then added to the medium for 1 h. Formazan crystals were dissolved using 100 µL of dimethyl sulfoxide (DMSO Sigma-Aldrich, Saint-Quentin-Fallavier, France). Optical density was evaluated at 570 nm using the Multiskan™ FC microplate photometer (Thermo Scientific™, Illkirch-Graffenstaden, France).

Cell proliferation: Cells were seeded in 96-well plates and incubated in the live cell imaging system IncuCyte^®^ (IncuCyte S3 Live-Cell Analysis System, Essen Bioscience, Ann Arbor, MI, USA). Cell confluency was measured and analyzed every 12 h over 96 h using the IncuCyte software. Experiments were performed in triplicate.

Wound Healing assay: cells were seeded at 3 × 10^5^ in 96-well plates (ImageLock™ plates, Essen Bioscience, Ann Arbor, MI, USA) and incubated until confluence. Wounds were homogenously generated manually using the 96-well Wound Maker (Essen Bioscience, Ann Arbor, MI, USA). Pictures were collected every 2 h during 24 h using IncuCyte^®^ apparatus before wound widths were analyzed. Results are expressed as wound confluency, which is the ratio of the occupied area of the initially scratched area to the total area of the scratch.

### 4.12. Subcutaneous Xenografts

Subcutaneous injection of CAPAN-1 LV-miR-neg or LV-miR-210 cells (10^6^ cells in 100 μL of Matrigel (R & D)) was performed into the flank of 8-week-old male severe combined immunodeficient mice (CB17, Charles Rivers, France). Groups consisted of six mice. Tumor volume was monitored by measuring length (L) and width (W) using the formula (W^2^ × L). All procedures were evaluated by the animal care committee (Comité Ethique Expérimentation Animale Nord Pas-de-Calais, #14123-2018012517309750).

### 4.13. Immunohistochemistry

Hematoxylin and eosin staining were performed on 5µm tissue sections. Primary antibodies specific for mouse Muc4 (1G8, Santa Cruz Biotechnology Inc., Dallas, TX, USA) and human Ki-67 (NLC Ki-67p Novocastra) were used for immunostaining as described previously [20,23]. The PanINs area was measured in each section using ImageJ software (National Institutes of Health, Bethesda, MD, USA) with a specific macro and reported relative to the total tissue area. The intensity of immunostaining was evaluated as follows: weak (1), moderate (2), or strong (3). Percentage of PanIN-stained cells: 1 (0–25%), 2 (25–50%), 3 (50–75%), and 4 (75–100%). Total Muc4 staining score was calculated by multiplying the PanIN areas with intensity and percentage scores. The number of Ki-67-stained cells was measured using ImageJ software (version 1.53k, National Institutes of Health, Bethesda, MD, USA) and reported to the total cell count.

### 4.14. Statistical Analyses

At least three independent experiments were performed for every assay. ANOVA and Student’s t-test statistical analyses were performed using Graphpad Prism 4.0 software (Graphpad softwares Inc., La Jolla, CA, USA). *p* < 0.05 was considered as statistically significant.

## 5. Conclusions

In the present work, we showed for the first time the existence of a feedback regulation between MUC4 and miR-210-3p. We observed that both miR-210-3p and MUC4 expression levels are increased in pancreatic cancer. We deciphered the complex MUC4–miR-210-3p negative regulation loop in PDAC. Indeed, we observed that MUC4 activates miR-210-3p transcriptional expression via NF-κB pathway modulation and leads to miR-210-3p overexpression in pancreatic cancer cells and tissues. On the contrary, miR-210-3p inhibits MUC4 expression via its 3′ UTR in vitro. Finally, we demonstrated in vitro and in vivo anti-proliferative and anti-migratory effects of miR-210-3p in PDAC, suggesting a complex balance between oncogenic roles of MUC4 and anti-tumoral roles of miR-210-3p.

## Figures and Tables

**Figure 1 cancers-13-06197-f001:**
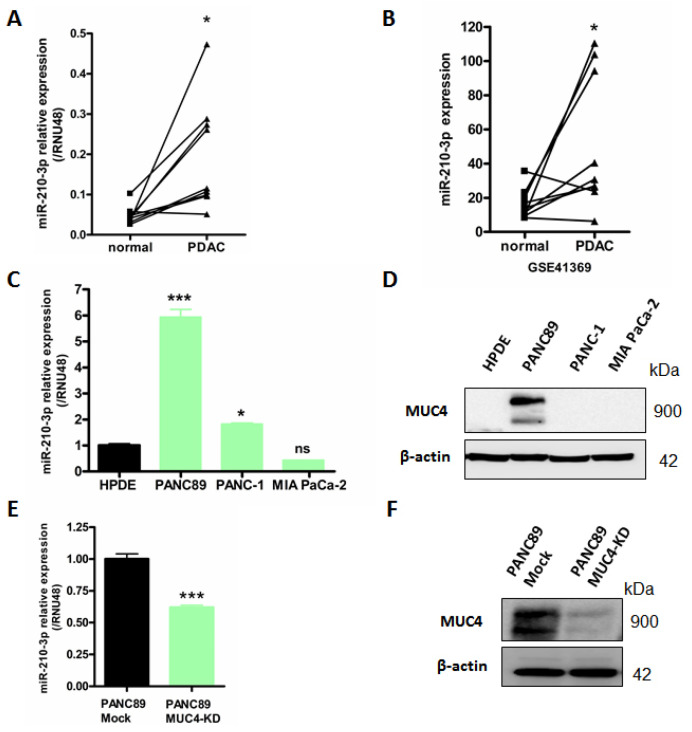
MiR-210-3p is overexpressed in PDAC tissues and pancreatic cancer cell lines compared to normal pancreas. (**A**) RT–qPCR analysis of miR-210-3p relative expression level in nine paired human pancreatic cancers and their corresponding non-tumoral adjacent tissues. Expression levels are evaluated using 2^−ΔCt^ method (ΔCt = Ct miR-210—Ct RNU48). (**B**) Analysis of miR-210-3p expression level in GSE41369 PDAC dataset using GEO2R analyzer. (**C**–**E**) RT–qPCR analysis of miR-210-3p relative expression in PANC89, PANC-1 and MIA PaCa-2 pancreatic cancer cells, HPDE normal human pancreatic ductal cells (**C**) and PANC89 Mock and MUC4-KD cells (**E**). Expressions were determined according to the 2^−ΔΔCt^ method (ΔΔCt = (Ct miR-210—Ct RNU48)—Ct HPDE). Three independent experiments were performed. (**D**–**F**) Western blotting analysis of MUC4 and β-actin expression in PANC89, PANC-1, MIA PaCa-2, HPDE (**D**) and Mock and MUC4-KD PANC89 cells (**F**). * *p* < 0.05 and *** *p* < 0.001 indicate statistical significance compared to normal tissues. ns indicates no statistical significance. At least three independent experiments were conducted.

**Figure 2 cancers-13-06197-f002:**
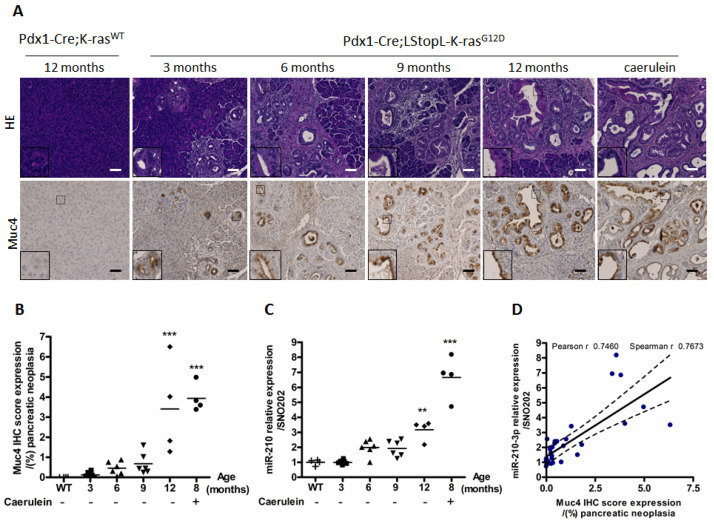
MiR-210-3p expression is correlated with Muc4 expression in PanIN lesions. (**A**) Hematoxylin and eosin stainings (upper panel) and Muc4 immunohistochemical staining (lower panel) were performed in pancreatic tissues from Pdx1-Cre; LStopL-K-rasG12D (KC) (3, 6, 9, 12 months and 8-month-old caerulein-treated mice) and Pdx1-Cre; K-rasWT (WT) control mice (12 months), representative of six mice per age. Scale bar: 100 µm. (**B**) An immunostaining score of Muc4 expression was calculated in PanIN lesions from KC mice treated, or not, with caerulein. *** indicates statistical significance (*p* < 0.001) compared to WT mice. (**C**) miR-210-3p relative expression level was evaluated by RT–qPCR in PanINs pancreatic tissue from KC mice treated or not with caerulein and K-ras WT control mice. Data are expressed according to the 2^−ΔΔCt^ method (ΔΔCt = (Ct miR-210-3p—Ct SNO202)—Ct K-rasWT). MiR-210-3p expression in WT control mice was arbitrarily set to 1. ** *p* < 0.01, *** *p* < 0.001 indicates a statistical significance compared to WT mice. (**D**) Correlation analysis between miR-210-3p expression and Muc4 expression in PanIN lesions. Statistical analyses were performed using Pearson’s and Spearman’s correlation coefficients (*r* = 0.7460, *r* = 0.7673, respectively, *p* < 0.0001).

**Figure 3 cancers-13-06197-f003:**
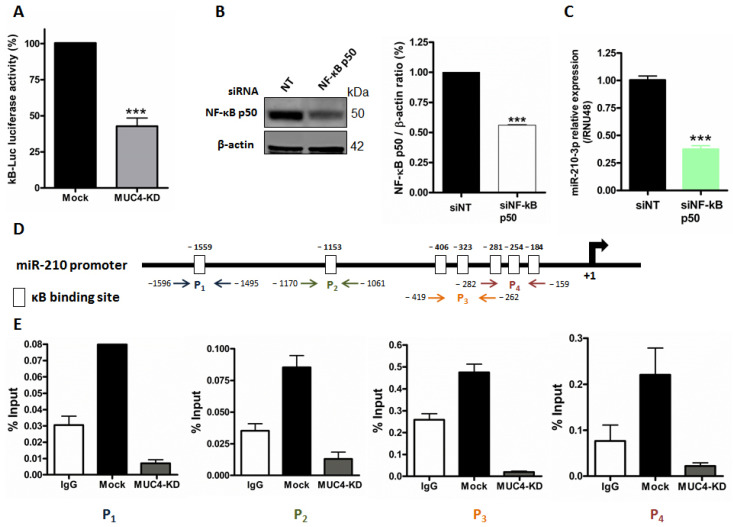
MUC4 regulates miR-210 expression through NF-κB pathway. (**A**) Luciferase activity of the κB-Luc synthetic reporter construct was measured 48 h following transient transfection in Mock and MUC4-KD PANC89 cells. Luciferase activity in Mock cells was set as 100%. *** indicates statistical significance compared to mock cells (*p* < 0.001). Three independent experiments were performed. (**B**) Western blotting analysis of NF-κB p50 and β-actin expression in siNF-κB p50 transfected cells and corresponding siNT control. P50/β-actin ratio is indicated on the histogram. (**C**) RT–qPCR analysis of miR-210-3p relative expression in NF-κB siRNA (siNF-κB p50)-transfected PANC89 cells. Expression in siNT control cells was arbitrarily set to 1. *** indicates statistical significance compared to siNT control cells (*p* < 0.001). Three independent experiments were performed. (**D**) miR-210 promoter schematic representation. κB predicted binding sites are indicated with white boxes. Four primer pairs (*p*) were used for ChIP assays and are indicated with colored arrows. (**E**) Chromatin immunoprecipitation of miR-210 promoter regions (P1[−1596/−1496], P2[−1070/−1061], P3[−419/−262] and P4[−282/−159] with NF-κB p50. Chromatin enrichment was normalized to the input. Three independent experiments were performed.

**Figure 4 cancers-13-06197-f004:**
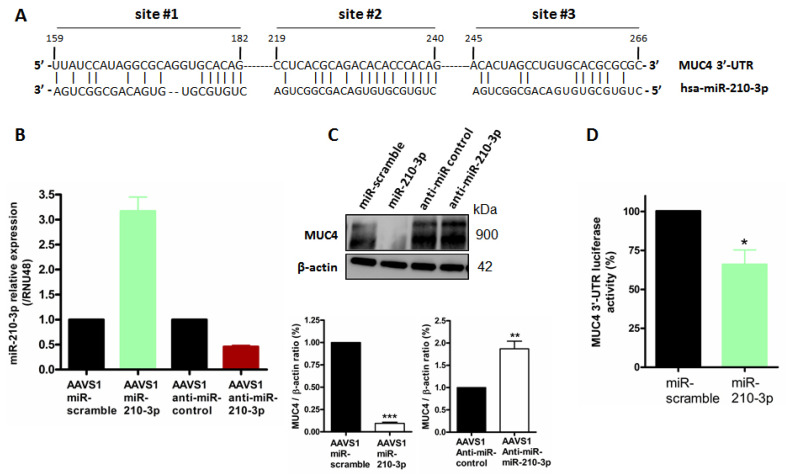
MiR-210-3p represses MUC4 expression via its 3′-UTR. (**A**) Identification of three putative miR-210-3p binding sites in *MUC4* 3′-UTR at the positions 219–240 (site #1), 159–182 (site #2) and 248–268 (site #3). (**B**) RT–qPCR analysis of miR-210-3p relative expression level in PANC89 stably transfected with miR-210-3p and anti-miR-210-3p and their respective controls, miR-scramble and anti-miR-control, using CRISPR/Cas9 recombination at AAVS1 integration site. Expression levels are evaluated with the 2^−ΔΔCt^ method (ΔΔCt = (Ct miR-210—Ct RNU48)—Ct miR-scramble or miR-control). (**C**) Western blotting analysis of MUC4 and β-actin expression in PANC89 AAVS1-miR-210-3p, PANC89 AAVS1-antimiR-210-3p and their controls PANC89 miR-scramble and PANC89 miR-control. The density of each band was measured by imageJ and control density was arbitrarily set to 1. MUC4/β-actin ratios are indicated on the histograms. Three independent experiments were performed. (**D**) PANC-1 cells were co-transfected with the pGL3-MUC4-3′-UTR reporter plasmid and the miR-210-3p or miR-scramble. Luciferase activity was measured 48 h after transfection and set as 100% in miR-scramble transfected cells. * *p* < 0.05 indicates statistical significance compared to miR-scramble cells (*p* = 0.0215). ** indicates *p* < 0.01; *** indicates *p* < 0.001. Three independent experiments were performed.

**Figure 5 cancers-13-06197-f005:**
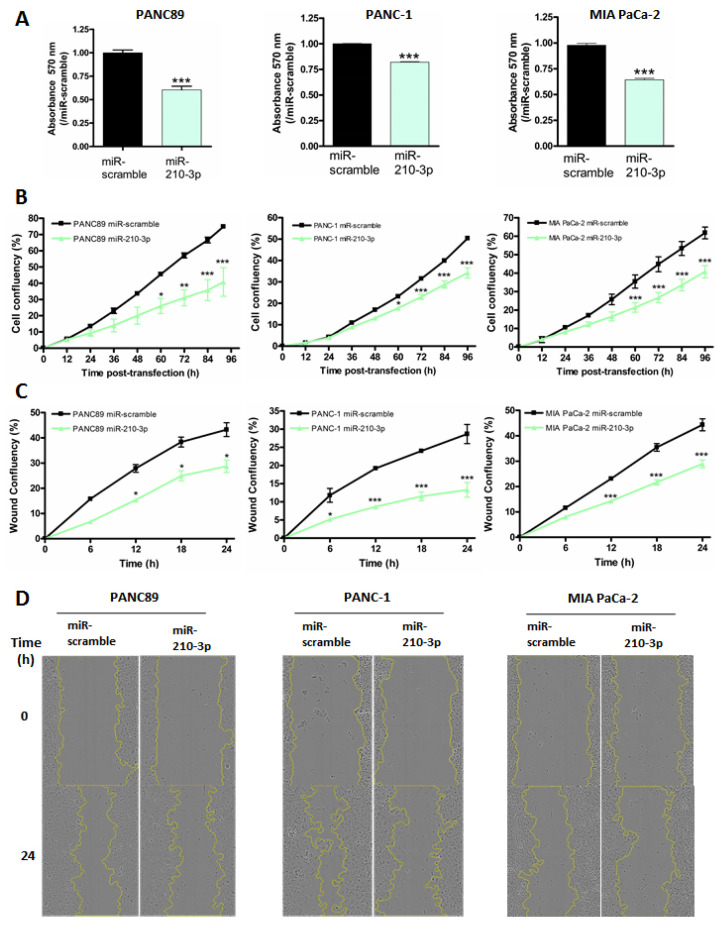
Transient expression of miR-210-3p inhibits proliferation and migration of pancreatic cancer cells. (**A**) PANC89, PANC-1 and MIA PaCa-2 cells were transiently transfected with miR-210-3p or miR-scramble for 96 h. Cell viability was assessed using MTT assays. MiR-scramble condition was arbitrarily set to 1. Three independent experiments were performed. (**B**) Cell confluency was evaluated using the Incucyte™ apparatus over 96 h following miR-210-3p or miR-scramble transient transfection of PANC89, PANC-1 and MIA PaCa-2 cells. Three independent experiments were performed. (**C**) Wound-healing assays were performed in transient transfected miR-210-3p or miR-scramble cells. Wound area was measured every 6 h for 24 h using an Incucyte™ instrument. Experiments were performed in triplicate. * *p* < 0.05, ** *p* < 0.01 and *** *p* < 0.001 indicate statistical significance compared to miR-scramble. Three independent experiments were performed. (**D**) Representative images of wound healing in PANC89, PANC-1 and MIA PaCa-2 cells.

**Figure 6 cancers-13-06197-f006:**
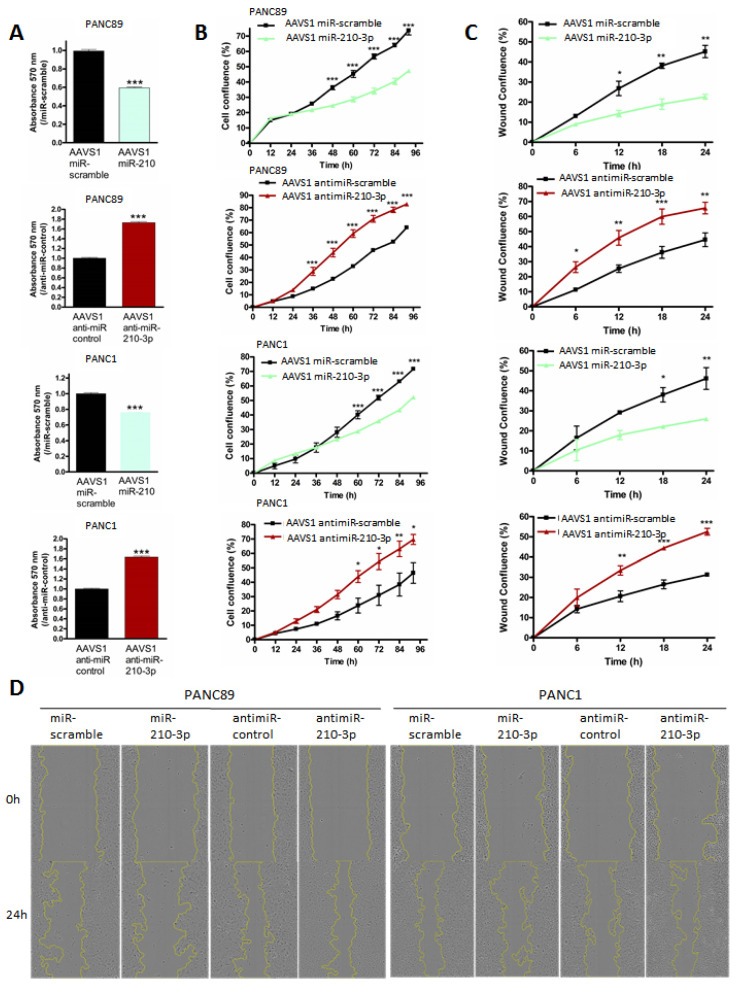
Stable expression of miR-210-3p and anti-miR-210-3p regulate the proliferation and migration of pancreatic cancer cells. PANC89 and PANC1 cells were stably transfected with either miR-210-3p, anti-miR-210-3p or miR-scramble and anti-miR-scramble corresponding controls using CRISPR/Cas9 recombination at the AAVS1 integration site. (**A**) Cell viability was assessed using MTT assays. MiR-scramble or anti-miR-scramble conditions were arbitrarily set to 1. Three independent experiments were performed. (**B**) Cell confluency was evaluated using the Incucyte™ instrument every 12 h during 96 h. Three independent experiments were performed. (**C**) Wound-healing assays were performed after seeding 30,000 cells. Wound area was measured every 6 h during 24 h using the Incucyte™ instrument. Three independent experiments were performed. (**D**) Representative images of wound healing in PANC89 and PANC-1 cells expressing miR-210-3p or anti-miR-210-3p. * *p* < 0.05, ** *p* < 0.01 and *** *p* < 0.001 indicate statistical significance compared to miR-scramble or anti-miR-control.

**Figure 7 cancers-13-06197-f007:**
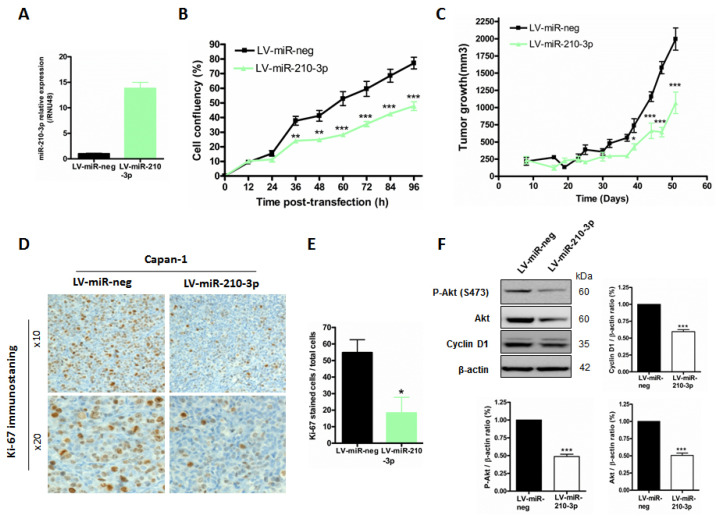
MiR-210-3p inhibits pancreatic tumor growth in vivo. (**A**) RT-qPCR analysis of miR-210-3p relative expression in Capan-1 LV-miR-210 and Capan-1 LV-miR-neg control cells. MiR-210-3p expression is evaluated according to the 2^−ΔΔCt^ method (ΔΔCt = (Ct miR-210—Ct RNU48)—Ct miR-neg). MiR-210-3p expression in Capan-1 LV-miR-neg control cells was arbitrarily set to 1. (**B**) Proliferation of Capan-1 LV-miR-210 cells compared with their controls, LV-miR-neg (** *p* < 0.01 or *** *p* < 0.001). Three independent experiments were performed. (**C**) Subcutaneous xenografts of Capan-1 LV-miR-210-3p and Capan-1 LV-miR-neg control cells in *SCID* mice. Tumor growth (mm^3^) was evaluated until mice were euthanized. * *p* < 0.05 and *** *p* < 0.001 indicate statistical significance of miR-210-3p compared with miR-neg control. (**D**) Ki-67 immuno-histochemical staining was performed on xenografted tumors. Original magnification: ×10 (upper panel) and ×20 (lower panel). (**E**) Ki-67^+^ percentage was calculated and represented as histograms. * *p* < 0.05 indicates statistical significance of miR-210-3p compared to miR-neg control. (**F**) Western blot analysis of S474 phospho-Akt, Akt, Cyclin D1 and β-actin protein expression in Capan-1 LV-miR-210-3p and Capan-1 LV-miR-neg control cells. Akt/β-actin, pAKT/β-actin, and Cyclin D1/β-actin ratios are indicated on the respective histograms. Three independent experiments were performed. *** *p* < 0.001 indicates statistical significance compared to Capan-1 LV-miR-neg control cells.

**Figure 8 cancers-13-06197-f008:**
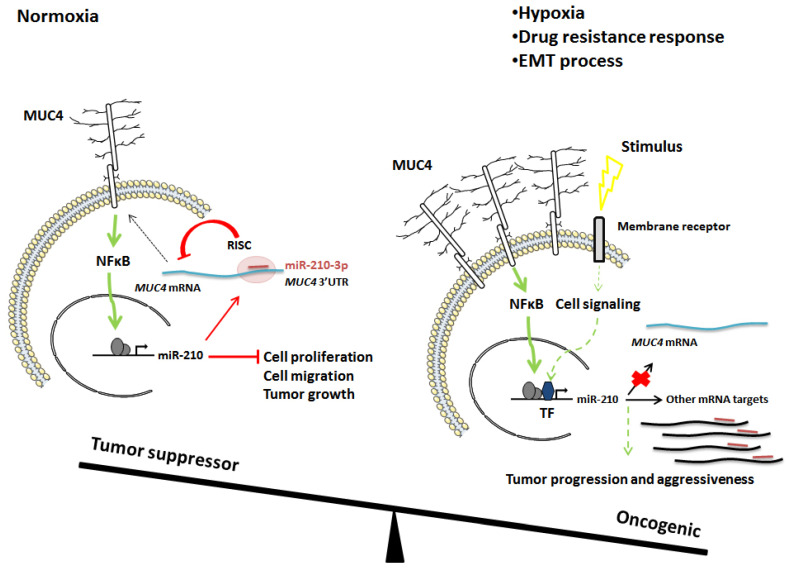
Schematic representation of overall miR-210-3p tumor suppressor/oncogenic effects in PDAC cells. Left: MUC4-miR-210-3p negative feedback loop that we identified under normoxic conditions. Right: Hypothetical model of miR-210-3p regulation and roles under hypoxia, drug resistance or metastasis signaling. Green arrows represent activating signals. Red arrows represent inhibitor signals. Short red line represents miR-210-3p. RISC: RNA-induced silencing complex. TF: transcription factors.

**Table 1 cancers-13-06197-t001:** Sequences of DNA blocks used for CRISPR/Cas9 cloning.

Block	Sense	Sequence (5′ → 3′)
miR-scamble	Forward	5′-CCGGTACACCATGTTGCCAGTCTCTAGGTGGGCGTATAGACGTGTTACACTGTGAAGCCACAGATGTGTAACACGTCTATACGCCCATGGCGTCTGGCCCAACCACACTTTTTG-3′
	Reverse	5′-AATTCAAAAAGTGTGGTTGGGCCAGACGCCATGGGCGTATAGACGTGTTACACATCTGTGGCTTCACAGTGTAACACGTCTATACGCCCACCTAGAGACTGGCAACATGGTGTA-3′
miR-210	Forward	5′-CCGGTACACCATGTTGCCAGTCTCTAGGAGCCCCTGCCCACCGCACACTGTGTGAAGCCACAGATCTGTGCGTGTGACAGCGGCTGATGGCGTCTGGCCCAACCACACTTTTTG-3′
	Reverse	5′-AATTCAAAAAGTGTGGTTGGGCCAGACGCCATCAGCCGCTGTCACACGCACAGATCTGTGGCTTCACACAGTGTGCGGTGGGCAGGGGCTCCTAGAGACTGGCAACATGGTGTA-3′
Anti-miR-control	Forward	5′-CCGGTAGAGCTCCCTTCAATCCAAGTGAAGAGCTCCCTTCAATCCAACGCGTAGAGCTCCCTTCAATCCAAATCGAGAGCTCCCTTCAATCCAACGCGTAGAGCTCCCTTCAATCCAAGTGAAGAGCTCCCTTCAATCCAACGCGTAGAGCTCCCTTCAATCCAAATCGAGAGCTCCCTTCAATCCAATTTTTG-3′
	Reverse	5′-AATTCAAAAATTGGATTGAAGGGAGCTCTCGATTTGGATTGAAGGGAGCTCTACGCGTTGGATTGAAGGGAGCTCTTCACTTGGATTGAAGGGAGCTCTACGCGTTGGATTGAAGGGAGCTCTCGATTTGGATTGAAGGGAGCTCTACGCGTTGGATTGAAGGGAGCTCTTCACTTGGATTGAAGGGAGCTCTA-3′
Anti-miR-210-3p	Forward	5′-CCGGTTCAGCCGCTGTCACACGCACAGGTGATCAGCCGCTGTCACACGCACAGCGCGTTCAGCCGCTGTCACACGCACAGATCGTCAGCCGCTGTCACACGCACAGCGCGTTCAGCCGCTGTCACACGCACAGGTGATCAGCCGCTGTCACACGCACAGCGCGTTCAGCCGCTGTCACACGCACAGATCGTCAGCCGCTGTCACACGCACAGTTTTTG-3′
	Reverse	5′-AATTCAAAAACTGTGCGTGTGACAGCGGCTGACGATCTGTGCGTGTGACAGCGGCTGAACGCGCTGTGCGTGTGACAGCGGCTGATCACCTGTGCGTGTGACAGCGGCTGAACGCGCTGTGCGTGTGACAGCGGCTGACGATCTGTGCGTGTGACAGCGGCTGAACGCGCTGTGCGTGTGACAGCGGCTGATCACCTGTGCGTGTGACAGCGGCTGAA-3′
SgRNA AAVS1		5′-GGGGCCACTAGGGACAGGATTGG-3′

**Table 2 cancers-13-06197-t002:** Primer sequences for ChIP experiments.

Position from TSS	Orientation	Sequences (5′ → 3′)
−159/−282	Forward	5′-GACCACCTCGGGCCGTACCAT-3′
	Reverse	5′-CTTTTCTGCACGTCTGCCCG-3
−262/−419	Forward	5′-CGGGAAGAGGGGCAGCTC-3′
	Reverse	5′-ATGGTACGGCCCGAGGTGGTC-3′
−1061/−1170	Forward	5′-CATGGGCTGGTTCGGAAGCTC-3′
	Reverse	5′-CATGACCTCCCTGCCTCGG-3′
−1475/−1596	Forward	5′-GGTGCCTGTGAAATTGGCAGGAC-3′
	Reverse	5′-GGGACAAGAAGGGGCAAGAGGAC-3′

## Data Availability

The data presented in this study are available on request from the corresponding authors.

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
