# Peer review of "Antagonistic Roles of the Tumor Suppressor miR-210-3p and Oncomucin MUC4 Forming a Negative Feedback Loop in Pancreatic Adenocarcinoma"

_cancers, 2021, doi:10.3390/cancers13246197_

Round 1

Reviewer 1 Report

The authors demonstrated the reciprocal role of miR-210 in relation to MUC4, which is known to be associated with disease progression and chemoresistance. The authors have well summarized the introduction section giving the readers the idea of the miR-210 and MUC4. The methods and results are well-organized but they have shown results that are opposing each other. This is the major point that should be further clarified. The up-regulation of MUC4  in the presence of high miR-210-3p expression was shown first but later, they have demonstrated the results of negative relation of miR-210 and MUC4. These results seem controversial and opposing each other. The authors merely suggest hypothesis that miR-210 negatively regulate MUC4 under normoxic conditions, however this should be supported by the data (measurement of oxygen levels in relation to miR-210 expression levels, development of hypoxia condition models and compare miR-210 expression levels), considering that this is the major statement of this manuscript. I don't quite agree with the authors that miRNA acts differently by the oxygen status and there are scarce information about miR-210 roles in normoxia. 

Reviewer 2 Report

Review recommendations to_Cancers-1457794-peer-review

The manuscript entitled “Antagonistic roles of the tumor suppressor miR-210-3p and oncomucin MUC4 forming a negative feedback loop in pancreatic adenocarcinoma”, reports on original research results conducted on pancreatic cancer cell lines, patient tissues and murine model of pancreatic cancers. The work reports biological roles of the oncogene, mucin4 and the tumor suppressor microRNA, miR-210-3p and their reciprocal regulation of tumor suppression and oncogenic effects via the involvement of NF-kB. The work is well designed and implemented, and the findings are well presented and expand the knowledge on the oncomucin MUC4 and miR-210-3p in carcinogenesis and drug resistance.

General comments:

-The writing style should be improved for clarity for a broad spectrum of readers not directly engaged in the same field.

-Placing the conclusions directly after the discussion section could be more appealing for readers.

Minor corrections recommended:

 Introduction:

Results:

  • Lines 231: please use “…overexpressing cells…”

Section 2.6:

  • Insert text before Figure 7. In the text change Figure A7 to 7A on line 312.
  • Why was Capan-1 used for in vivo studies but not included in the detailed in vitro studies? Does Capan-1 express high MUC4 levels?

Materials and methods:

  • Lines 425, 432 and 437: Change “included in paraffin” to “embedded in paraffin”.
  •  
  • Section 4.3: Please insert information on the Capan-1 cell lines here and their culture conditions. Since the bladder cancer cell line 647-V was also used and described here, change the title of this section to “Cell lines used and culture conditions”.
  •  
  • Line 546: correct Peroxidase.
  • Line 578: delete “were” from “…was were…”

Conclusions: The results are very good but the conclusions do not portray this strong enough. Also insert this directly after the discussions.

Supplementary materials

  • I suppose inserting only the titles of the Supplementary Figures in the main section in the manuscript is enough. Since the figures carry detailed legends.

Reviewer 3 Report

The authors present a well written study that tries to demonstrate a MUC4-miR-210-3p negative feedback loop in early onset of PDAC, and new functions of miR-210-3p in both in vitro and in vivo proliferation and migration of pancreatic cancer cells. Moreover, this article could suggest a complex balance between MUC4 pro-oncogenic roles and miR-210-3p anti-tumoral effects.

The reviewer feels that this study could be published after addressing several major and minor concerns.

Major concerns:

In the section 2.1 the Authors showed that MiR-210-3p is overexpressed in PDAC. Moreover, the Authors in the section 2.2 highlight a positive correlation between MUC4 and miR-210- 3p expression levels in mice and patients PDAC samples and suggest a potential MUC4 implication in miR-210-3p regulation. On the other hand, the authors showed that in PDAC-derived cells miR-210-3p represses MUC4 expression via MUC4 3’-UTR.

Therefore, the repression of MUC4 expression mediated in vitro by miR-210- 174 3p is in disagreement with in vivo positive correlation between MUC4 and miR-210- 3p expression. The authors must explain this point.

In the section 2.6 the authors showed results concerning the pancreatic tumour growth inhibition  mediated by miR-210- 174 3p. In particular, in the Figure 7 F the authors should  also analyze levels of pAKT that regulates cancer growth (PMID: 30905820 DOI: 10.1016/j.canlet.2019.03.017). Moreover, the Authors should add a histogram representing densitometric analysis of three independent experiments, with p value.

Minor concerns

Figure 1. The authors should define the meaning of "*" and "***" by adding their p value in the caption. For instance "*p<0.05, **p<..., ***p<..., etc.".

Figure 1C. The authors should decorate the graph of figure1C with asterisks to denote statistical significance among the samples. Furthermore, the authors should specify the significance level of the asterisk/s also in the caption.

Figure 1D-F. The authors should specify how many independent experiments were performed for the evaluation of MUC4 protein expression by Western Blotting.

Figure 2° The authors should add to hematoxylin and eosin staining, Muc4 immunohistochimical staining scal bars and magnifications.

In figures 3B, the authors should report the number of independent experiments performed to evaluate NF-κB p50 and β-actin protein expression. Furthermore, line 186, the authors should add the p value of data of the western blot analyses.

The results of wound healing analyses, showed in figures 5,6, are not so clear and direct.

Therefore, to help make these data cleaner and more readable, the authors should report the formula used to calculate the values of "wound confluency" or indicate these wound data as percentage of fold-decrease of "open wound area" compared with the wound width at time 0h (see Materials and Methods DOI:10.3390/cells8111435).

Lines 223-226, the authors should report p value of the data presented.

Lines 322-323, the authors should specify if the decrease of Akt and Cyclin D1 protein expression is statistically significant. In that case, the authors should report the p value.

Correct typos. For instance: line264, change "cell-migration" with "cell migration"; line 332 change “anti-tumor miRNA” with anti-tumor miRNA”; line 380, add comma after “methylation and histone modifications [34]”.

Round 2

Reviewer 1 Report

No further comments. 

Author Response

We thank the reviewer for insightful comments that really improved the manuscript.

Reviewer 3 Report

The authors have significantly improved the manuscript. There are only some minor concerns that should be addressed before publication:

Figure 1. The authors should change the sentence “*p<0.05 indicates statistical significance compared to normal tissues. *** p<0.001 “ with “*p<0.05 and  *** p<0.001   indicate statistical significance.”

The authors should specify in the legend what NS means (Figure 1 C).

Figure 7. The authors should change the sentence “Akt/β-actin and Cyclin D1/β-actin ratios are indicated on the respective histograms” with” pAKT/ β-actin, Akt/β-actin and Cyclin D1/β-actin ratios are indicated on the respective  histograms”.

Author Response

We corrected every minor request as requested (highlighted using red text in Figure 1 and Figure 7 legends)